# Dietary Antiplatelets: A New Perspective on the Health Benefits of the Water-Soluble Tomato Concentrate Fruitflow^®^

**DOI:** 10.3390/nu13072184

**Published:** 2021-06-25

**Authors:** Niamh O’Kennedy, Ruedi Duss, Asim K Duttaroy

**Affiliations:** 1Provexis PLC, c/o The University of Aberdeen, Polwarth Building, Foresterhill, Aberdeen AB25 2ZD, UK; niamh.okennedy@provexis.com; 2DSM Nutritional Products Ltd., 4002 Basel, Switzerland; Ruedi.Duss@dsm.com; 3Department of Nutrition, Institute of Basic Medical Sciences, Faculty of Medicine, University of Oslo, 0315 Oslo, Norway

**Keywords:** platelet function, platelet activation, platelet hyperactivity, platelet-leukocyte aggregates, inflammation, immunothrombosis, particulate air pollution, dietary antiplatelet, water-soluble tomato extract

## Abstract

Our understanding of platelet functionality has undergone a sea change in the last decade. No longer are platelets viewed simply as regulators of haemostasis; they are now acknowledged to be pivotal in coordinating the inflammatory and immune responses. This expanded role for platelets brings new opportunities for controlling a range of health conditions, targeting platelet activation and their interactions with other vascular cells. Antiplatelet drugs may be of wider utility than ever expected but often cause platelet suppression too strong to be used out of clinical settings. Dietary antiplatelets represent a nutritional approach that can be efficacious while safe for general use. In this review, we discuss potential new uses for dietary antiplatelets outside the field of cardiovascular health, with specific reference to the water-soluble tomato extract Fruitflow^®^. Its uses in different aspects of inflammation and immune function are discussed, highlighting exercise-induced inflammation, mediating the effects of air pollution, and controlling thrombotic aspects of the immune response. Potential future developments in women’s health, erectile dysfunction, and the allergic response indicate how broad the utility of dietary antiplatelets can be.

## 1. Introduction

Platelets play a key role in maintaining homeostasis of the blood and preserving the integrity of the vascular system. These anuclear cell fragments possess a varied range of cell surface receptors and produce many signalling molecules released on activation [1]. Their ability to respond to a diverse array of external stimuli renders them exquisitely sensitive to any stresses experienced by the vascular system. However, this sensitivity can also result in a hyperaggregable state, where a proportion of circulating platelets are chronically sensitised by their environment—the blood and vascular endothelium—and can contribute to pathologic processes [1,2]. The most widely studied example is platelet influence on the development and progression of cardiovascular disease (CVD); antiplatelet therapy has been a cornerstone of treatment for established CVD for many years. Platelets are also involved in immune responses and host defences, and recent work has suggested that the haemostatic and immune systems are intimately linked rather than separate entities [3]. With growing recognition of the critical role of platelets in inflammation and immune responses [4], it has become evident that control of platelet-driven inflammatory responses in many diseases other than CVD is important. Recent studies have indicated that antiplatelet medications may reduce mortality from infections and sepsis [5].

While antiplatelet medications are appropriate for the treatment of illness or control of disease, they are unsuitable in the context of primary prevention. Antiplatelet drugs—for example, acetylsalicylic acid, dipyramidole, and abciximab—typically act on a particular pathway of platelet aggregation, which, as a result, is completely suppressed. This level of platelet suppression is linked to increased bleeding tendencies, to the extent that risk from bleeding outweighs the antiplatelet benefits. Nutritional approaches to lowering platelet hyperactivity are attractive, and quite a range of plant-derived antiplatelets are known, including pycnogenol from pine bark [6], sulfur derivatives from garlic [7], flavonoids from onion, grapes and olive oil, and berry anthocyanidins are some of the most well studied [8]. Some berry extracts, such as Aronox (available from Agropharm LTD, Tuszyn, Poland), an extract from *Aronia melanocarpa*; and Delphinol (available from Maqui New Life, Santiago, Chile), an extract from the maqui berry *Aristotelia chilensis*, are commercially available. Although they are marketed mainly as antioxidants, they show some antiplatelet properties [9]. However, there is little coherent data on the efficacy of using these dietary antiplatelets.

The water-soluble tomato extract known as Fruitflow^®^ is a dietary antiplatelet. Fruitflow^®^ contains a range of tomato-derived secondary metabolites, including nucleosides, phenolic conjugates, and polyphenols, all of which show different profiles of antiplatelet activity [10]. The compounds present are all water-soluble and of low molecular weight; some (for example, nucleosides) are absorbed from the upper gastric tract very quickly upon ingestion, whereas others (for example, flavonoid glycosides) are absorbed later in the digestive process, either from the lower small intestine or after microbial interaction in the proximal colon. The different metabolic fates of the antiplatelet compounds result in antiplatelet effects, which are observable 1.5–3 h after ingestion and persist (from a single dose) for around 18 h [10]. When consumed daily, the effects are continuous [11]. Fruitflow^®^ reduces platelet aggregation in response to major platelet agonists, such as adenosine diphosphate (ADP), collagen, arachidonic acid, and thrombin [10,12]. A range of human studies in healthy subjects has established an average reduction in platelet aggregation by 9–23%, depending on the agonist used [10,12,13,14,15]. Studies have been conducted in both healthy subjects and some patient groups, particularly in hypertensives [13,16]. In Europe, the level of platelet suppression observed has been judged suitable for use in primary prevention of platelet hyperactivity, and a specific health claim has been granted: Fruitflow^®^ helps maintain normal platelet aggregation, which contributes to healthy blood flow [17].

We previously summarised the body of work related to Fruitflow^®^’s effects on platelet aggregation, most of which was conducted in the context of cardiovascular health [11]. This review focuses on potential uses for a standardised dietary antiplatelet such as Fruitflow^®^ in the areas of inflammation and immune function and suggests other potential targets for future research into the health benefits of dietary antiplatelets.

## 2. Overview of the Active Components, Mechanisms of Action, Efficacy and Safety of Fruitflow^®^

Since the composition, preparation, and antiplatelet effects of Fruitflow^®^ have been reported elsewhere [10,11,18,19,20,21,22,23,24], we include a brief description of the ingredient and its main mechanisms of action in this study. Fruitflow^®^ is derived from ripe tomatoes using a process that removes pomace, seeds, and fats while operating at low processing temperatures (~40 °C) to control the production of artefacts such as Maillard/Amadori products or Strecker aldehydes. Thus, it is a lycopene-free water-soluble extract with a compositional profile as close to fresh clarified tomato juice as practical. It is further processed to remove both polysaccharides and soluble sugars, which comprise most of its dry matter content but does not contribute to its antiplatelet activity. The resulting powder contains a range of nucleotides and nucleosides (including adenosine, cytidine, guanosine, AMP, GMP, and deoxy derivatives), a range of simple phenolic compounds (e.g., caffeic and ferulic acids, glycosides, and conjugates with quinic acid) and a range of flavonoid derivatives in which quercetin derivatives dominate. The processing conditions are controlled to prevent degradation of glycosides and other derivatives, and to retain a compositional profile close to that of fresh tomato juice. Fruitflow^®^ is standardised regarding representatives of its three main classes of antiplatelet compounds—nucleosides, phenolic derivatives, and flavonoid derivatives—and the total amount of ‘bioactive extract’ present. It is also standardised by bioassay, measuring its IC50 in preventing platelet aggregation in response to ADP, collagen, arachidonic acid, and thrombin. The production of a standardised extract that can be consumed in small amounts has facilitated research into the antiplatelet effects of tomato compounds, avoiding the variability inherent in using juices or other food formats.

Proteomic experiments conducted to examine the effects of Fruitflow^®^ on platelet signalling pathways have shown that those regulating platelet structure, coagulation and redox status are strongly affected. The downstream effects of changes to these signalling pathways are observed in reduced activation of integrin αIIbß3 (GPIIb/IIIa), an activation step common to multiple aggregation pathways, as well as lower induction of P-selectin (CD62P) on the platelet surface, and lower binding of circulating tissue factor (TF). These changes functionally alter platelet capacity to form stable aggregates and to activate thrombin generation. Effects on protein disulphide isomerase (PDI), an oxidoreductase that catalyses the formation and the isomerisation of disulfide bonds, may underly many of these changes to platelet function and be linked specifically to quercetin derivatives present. In platelets, blocking PDI with inhibitory antibodies inhibits several platelet activation pathways, including aggregation, secretion, and fibrinogen binding.

The effects of Fruitflow^®^ in humans was demonstrated in ten studies focused on platelet aggregation and hypertension (see review [11]). On average, these studies showed an inhibition of the platelet response to ADP agonist by approximately 9–23% and inhibition of the response to collagen by approximately 10–18%. Arachidonic acid-induced and thrombin receptor-activating peptide (TRAP)-induced platelet aggregation were shown to fall after Fruitflow^®^ administration. A study in which Fruitflow^®^ was administered to 93 healthy men and women showed that some variability in response might occur, with men more responsive than women and subjects with higher risk factors for CVD having higher responsiveness than others. A dose-response was established in studies administering different amounts of Fruitflow^®^. As compounds found in Fruitflow^®^ have been shown to affect many aspects of platelet function, including (via effects on TF immobilisation and signalling) thrombin generation, during all human intervention studies, care was taken to incorporate specific safety-focused measures to examine whether any effects on the intrinsic or extrinsic clotting pathways could be detected alongside antiplatelet effects. An antiplatelet which also affects blood coagulation pathways can raise safety concerns. However, in all intervention studies undertaken, clotting time measurements showed no significant increase from baseline levels. Fruitflow^®^ does not directly affect blood coagulation at any dose tested. Antiplatelet effects were placed into clinical context by comparing the effects of Fruitflow^®^ with those of 75 mg aspirin; in this study, it was shown that Fruitflow affects platelets approximately one third as strong as aspirin and does not dangerously elevate the bleeding time in the manner that aspirin does [14]. The lower strength of its effects on the haemostatic system, compared with aspirin, can be linked to the fact that Fruitflow does not have a cumulative effect when consumed daily, unlike aspirin [14]. Thus, the efficacy and safety profile of Fruitflow^®^ are suited to use in primary prevention, whereas aspirin is not.

## 3. Dietary Antiplatelets in Inflammation

Platelets release a plethora of inflammatory mediators with no known role in haemostasis. Many of these mediators modify leukocyte and endothelial responses to a range of different inflammatory stimuli [25]. Platelet-leukocyte aggregates are now regarded as a key aspect of the inflammatory response; bridges between leukocytes and endothelium are largely mediated by platelet P-selectin [25,26]. Platelets have emerged as crucial coordinators of inflammation through their interactions with monocytes, neutrophils, lymphocytes, and the endothelium. As a response to injury or disease, the versatility and reactivity of platelets in recruiting leukocytes and initiating an inflammatory response is highly beneficial. However, the reactivity of platelets also brings disadvantages, sometimes generating and maintaining a raised inflammatory burden which accelerates tissue damage or the progress of a disease, for example, in atherosclerosis, diabetes or inflammatory bowel disease [27,28,29]. In such conditions, the use of antiplatelet therapy is common to restrict platelet hyperactivation and control inflammation.

Dietary antiplatelets become operative under circumstances where persistent platelet activation arises in response to dietary and lifestyle factors, most notably smoking [30] or exposure to smoke/air pollution [31], consistently high levels of plasma glucose [32], and certain patterns of exercise [33]. Some degree of platelet hyperactivity is common among apparently healthy subjects, and while this does not constitute a ‘health condition’ in itself, it results in interactions that increase the release of IL-6 and NFkB [25], increase the levels of circulating microparticles (most of which are platelet-derived) [34], increase thrombotic tendencies [35], and raise CRP levels [4,36]. The increased inflammatory burden can hinder the body’s response to external stresses and lower general resilience, as well as predisposing early development of conditions such as diabetes or atherosclerosis over time. Dietary antiplatelets have a clear role in modifying the platelet response to a pro-inflammatory environment in the vascular system when it is unsuitable to use antiplatelet drugs for this purpose. Two specific instances illustrating such usage will be discussed.

### 3.1. Exercise-Induced Inflammation

Frequent, moderate exercise is known to benefit health in a multitude of ways, not least in managing cardiovascular risk [37]. When undertaken as part of a healthy lifestyle in combination with a balanced diet, moderate exercise of various types improves both quality of life and overall health [38]. The premise that there may be benefits from, or even requirements for, antiplatelet action during exercise is often met with surprise. Exercise is associated with good health, and increasing exercise is usually linked to longer life, better mental health, less metabolic problems, stronger bones, and improved cardiovascular health [39]. However, strenuous exercise is associated with an increased risk of vascular thrombotic events and sudden death [33]. We experience stress related to exercise as lack of breath, muscular fatigue, or even acute pain. On a molecular level, during exercise, we are inducing an inflammatory burst, which is mediated by platelets. Strenuous exercise releases adrenalin and serotonin and generates thrombin [40], resulting in platelet activation [33]. In addition, intense aerobic exercise can reduce the amount of anti-aggregatory nitric oxide (NO) produced by the vascular endothelium, especially in untrained subjects, as the amount of oxygen reaching the NO-producing cells is reduced [41]. Hyperaggregability develops, and platelets then coordinate a series of pro-inflammatory events as described previously [42]. This sequence of events has two major consequences. Firstly, after exercise, the potential for blood to coagulate and form a blood clot increases. This state is termed hypercoagulability, and can last for up to 48 h after an exercise session [43]. The extent of hypercoagulability depends on the duration and intensity of the exercise undertaken as well as training status; it is worse in untrained individuals at lower intensities of exercise but still occurs in well-trained individuals at higher intensities of exercise. This increase in coagulation capacity can be dangerous, especially for those with underlying health conditions such as atherosclerosis or cardiac problems, leading to an increased risk of thrombosis and sometimes even sudden death [44]. It has been estimated that physically inactive individuals have an approximately 50-fold increase in the risk of sudden death and a 100-fold increase in the risk of a heart attack when they perform vigorous exercise [45]. These risk levels can be substantially reduced by regular physical training, but an association with vigorous exercise remains (risk raised two- to five-fold in athletes). Approximately 70% of exercise-induced sudden deaths and heart attacks in the over-35 age group are attributed to obstruction of arteries by platelet clots [45].

The second consequence of platelet activation during exercise is increased inflammation, manifesting as increased circulating IL-6, increased circulating microparticles, and leukocyte and reactive oxygen species (ROS) accumulation in muscle after exercise. These parameters are linked to longer recovery times [46]. Typical signs of this are delayed muscle soreness, muscle damage due to poor recovery, and difficulty adhering to a training regime.

Many studies have examined the potential for nutritional interventions to reduce the inflammatory markers mentioned above after exercise, but results are generally inconsistent. Most of the interventions used primarily target either IL-6 or ROS [47,48]. However, exercise-induced inflammation is partly driven by hypercoagulability, and this is governed by thrombin generation, increased TF expression, and (to a lesser extent) epinephrine release. Thus, using a dietary antiplatelet to decrease hypercoagulability is a rational hypothesis. Most dietary antiplatelets, however, do not directly affect thrombin generation or TF, and very few are documented to affect epinephrine-mediated aggregation. The exception is Fruitflow^®^, which reduces TF binding to P-selectin, thus reducing platelet-generated thrombin release. Fruitflow^®^ also directly reduces the platelet response to circulating thrombin. To examine the likely effect of a dietary antiplatelet in this area, we conducted preliminary experiments to examine the in vitro effects of Fruitflow^®^ on thrombin- and epinephrine-stimulated platelets and on human umbilical cord endothelial cells (HUVEC) stimulated by activated platelet-leukocyte suspension. Treatment of HUVEC cells with Fruitflow^®^ prior to stimulation reduced platelet aggregation and microparticle formation (index of platelet activation) by 91% and 31%, respectively, compared with control. Fruitflow^®^ treatment reduced IL-6 generation by stimulated HUVEC cells by approximately 80%, compared with non-stimulated control [49]. Thus mechanistically, it appears that Fruitflow, or another dietary antiplatelet affecting thrombin generation, has potential as an intervention in exercise-mediated hypercoagulability and the inflammation arising.

A small exploratory study using a treadmill test in six untrained men aged 18–55 showed that exercising for 20 min at 70% VO_2max_, followed by exercising at 90% VO_2max_ until self-determined exhaustion, resulted in a post-exercise increase in markers of platelet activation, coagulation and inflammation (Table 1). Consuming Fruitflow^®^ 90 min prior to exercise led to decreased systemic activation compared with a placebo supplement. Exercise-induced platelet microparticle generation was reduced by 77% compared with control, and the induced increase in thrombin generation capacity was reduced by 89%. The reduced haemostatic system activation was accompanied by reduced circulating IL-6, which was 58% lower post-exercise after consuming Fruitflow^®^ than after consuming a placebo.

The work discussed in this study is preliminary and serves only to illustrate the level of systemic activation which may arise after vigorous exercise. Larger studies designed with a suitable statistical power are needed to thoroughly investigate the effects of exercise on platelet-related parameters. The use of animal models (for example, mice models, including several knockouts) may also help examine the impact of factors such as degree of hypoxia or aging on the strength and underlying mechanisms of exercise-induced platelet activation [50]. It remains to be seen whether reducing induced inflammation can significantly impact subsequent muscle soreness and exercise frequency, thereby enabling a healthier and fitter lifestyle, which is especially important with increasing age. However, these exploratory experiments offer the interesting suggestion that dietary antiplatelets can be potentially useful in reducing exercise-induced inflammation where antioxidants have not proven efficacious.

### 3.2. Air Pollution

The majority of the world’s population (92%) currently breathe air that fails to meet World Health Organization guidelines. As reported by the Global Burden of Disease Report, outdoor fine particulate matter (particulate matter with an aerodynamic diameter <2.5 μm) exposure is the fifth leading risk factor for death in the world, accounting for 4.2 million deaths [51]. The World Health Organization attributes 3.8 million additional deaths to indoor air pollution. As such, air pollution is now the largest environmental risk factor for ill health. Pollutants in the air come from a range of sources and are defined as primary pollutants if released directly from industrial or transportation activities (e.g., sulfur, nitrogen oxides, and carbon monoxide), or secondary pollutants if formed in the atmosphere via interaction with primary pollutants (e.g., ozone, particulates). Arguably, the most toxic form of air pollution is particulate matter (PM).

Particulate air pollution is related to natural events—volcanic emissions, dust storms, forest fires—and human activities such as vehicle or machinery emissions and traditional cooking practices. Both types of events result in the suspension of soot, gases and other air particulate matter (PM). PM is usually classified by its size; PM_10_ denotes particles <10 μm in diameter, PM_2.5_ particles are <2.5 μm in diameter, and PM_0.1_ particles are <0.1 μm in diameter. Smaller PM is more toxic than larger PM, as they are easily transported to more tissues in the body. PM_2.5_ are small enough to penetrate lung alveoli, while PM_0.1_ pass through the alveolar-capillary membrane and into the bloodstream [51]. PM in the bloodstream induces cytotoxic and inflammatory responses, and there is a recognised link between exposure to diesel emissions and cardiovascular disease [52]. Particulate matter promotes arterial thrombosis and atherosclerosis through increased platelet activation [31], leading to accelerated coronary heart disease and strokes, which are the main causes of death from air pollution [51,52].

In a 2018 review on the impact of air pollution on thrombosis, Robertson et al. concluded that acute exposure to PM_2.5_ shifts the haemostatic balance towards a pro-thrombotic/pro-coagulative state [31]. Platelet activation and oxidative stress resulted in the formation of platelet-leukocyte conjugates and an increase in circulating IL-6. The interactions between platelets and TF appear to be relevant to activation in these conditions. A role for circulating platelet and leukocyte derived microparticles is also beginning to emerge [53]. Due to the chronic nature of air pollution exposure, epigenetic changes are likely to develop heritable pro-thrombotic genotypes [54].

As mentioned previously, Fruitflow^®^ exerts some of its antiplatelet effects through mechanisms involving suppression of P-selectin and concomitant suppression of TF binding to the platelet. It also reduces platelet microparticle formation [11]. Since both mechanisms are implicated in platelet activation by air pollution, we conducted some exploratory tests to expose platelets to airborne particulate matter, such as diesel emissions, in the presence or absence of Fruitflow^®^. These in vitro tests showed that Fruitflow^®^ reduces the platelet activation caused by PM_2.5_ by approximately one third [55]. Our early work showed that the phenolic glycosides contained in Fruitflow^®^ are strongly linked with their effects on thrombin-mediated platelet activation; we hypothesise that compounds such as chlorogenic acid and other caffeic acid glycosides, as well as ester derivatives of phenolic acids, are important in mediating these effects. Other dietary antiplatelets, such as phenolic fractions from olive oil, or onions, may also be of interest if standardized extracts become available.

As of yet, there is no coherent body of work examining the effects of nutrition on the damage caused by air pollution. Whyand et al. concluded that while some nutritional components may be associated with some benefits (e.g., Vitamin D, Vitamin E, carotenoids, omega-3 oils, and the Mediterranean diet), there is little direct evidence of specific protective effects, and more studies are needed urgently [56]. We suggest that dietary antiplatelets can be an interesting topic of investigation. Protecting platelets from the toxic effects of PM_2.5_ and PM_0.1_ may help reduce the inflammatory burden, which develops and worsens over time. However, the vast differences in individual genetic susceptibility and environmental conditions render exploring and validating this hypothesis a complex task. Several established animal models, particularly in rats, have been used to examine the mechanisms by which air pollution affects the cardiovascular and pulmonary systems [57]; such investigations may help to further elucidate the impact of platelet suppression on the systemic response to airborne particulate matter.

## 4. Dietary Antiplatelets in Immunity

As we discussed previously, platelets interact with circulating leukocytes in response to inflammatory stimulus primarily by inducing surface expression of P-selectin or the surface glycoprotein, CD40, which bind to leukocytes PSGL1 and CD154 (CD40L), respectively, leading to the formation of platelet–leukocyte aggregates [25]. When these aggregates are formed with circulating monocytes or neutrophils and trigger an inflammatory response, the innate immune response is triggered [4]. Interactions of platelets with innate and adaptive immune responses have been among the foremost areas of research in platelet biology in recent years; it is now well established that platelets have a pivotal role in initiating and modifying the immune response [58,59,60]. Platelets contain the mRNA transcripts for all TLR1 to TLR10 (Toll-like receptors). These molecular pattern recognition receptors are key regulators in initiating the innate immune response to foreign organisms [61,62]. Platelet mediated CD40-CD154 interactions take place not only with leukocytes but also with dendritic cells, leading them to present antigen to T cells [4]. In addition, platelet activation also leads to the release of δ-granule content and secretion of molecules such as serotonin and RANTES (regulated on activation; normal T cell expressed and secreted; CCL5) that are also known to mediate T-cell activation and differentiation. Platelets directly recognize and internalize pathogens [63], and modulate leukocyte behaviour, enhancing their ability to phagocytose and kill pathogens [64]. Platelets are also involved in coordinating the production of Neutrophil Extracellular Traps (NETs) [65].

During certain stages of infection, platelet-initiated activation of both innate and adaptive immunity is beneficial to the host. When tissue damage is caused by blood-borne pathogens (viral or bacterial), the multifaceted platelet response involving multiple cellular interactions and secretions results in a coordinated intravascular coagulation response termed immunothrombosis [66]. During this process, platelets and immune cells form a physical barrier of confinement, preventing the dissemination of pathogens. However, uncontrolled endothelial damage and inflammation resulting from infection progression can lead to adverse prothrombotic responses and increased cardiovascular risk [66]. For example, a thrombotic disease often associated with infection is septicemia, which leads to disseminated intravascular coagulation (DIC). DIC is characterized by microthrombus formation, blocking the microvasculature and causing widespread tissue/organ damage. Another example is the enhanced immunothrombosis that characterizes severe cases of COVID-19 disease caused by the SARS-nCoV-2 virus. Severe infections with SARS-nCoV-2 can result in a cytokine storm, systemic inflammatory response and immunothrombosis, leading to microvascular thrombosis (widespread blood clots in tiny blood vessels) [67].

Antiplatelet therapy is often used in cases of severe infection and some cases of less severe infection (e.g., use of acetylsalicylic acid in influenza). However, its use has become much more widespread since the advent of the SARS-nCoV-2 virus. Strong antiplatelet treatment is neither practical nor suitable for most COVID-19 patients who contract illness of mild or medium severity. However, targeting platelet hyperactivity may help prevent or delay the progression of illness from mild to severe. Studies recently published by Manne et al. [68] and Hottz et al. [69] showed that production of TF (and by extension, thrombin) was blocked in COVID-19 illness by targeting the platelet. This result was achieved by pretreating COVID-19 patient platelets with an anti-P-selectin neutralizing antibody or the clinically approved anti-αIIb/β3 monoclonal antibody, abciximab. The enhanced role of antiplatelets in managing COVID-19 has raised interest in nutritional interventions that might help manage the progress of COVID-19, and similar, infections. The potential for nutritional status to impact the progress of thrombotic complications in COVID-19 infection was reviewed in 2020 by Tsoupras et al. [70]. This article concluded that nutrients with antithrombotic properties could benefit individuals infected by the SARS-CoV2 virus.

The underlying platelet hyperactivity associated with conditions such as diabetes and heart disease, or found in areas of high pollution, is significant; the importance of addressing this issue has been laid bare by the consequently higher risk of thrombosis observed in COVID-19 infection. We suggest that a dietary antiplatelet such as Fruitflow^®^ be considered for daily use, especially in those known to have a higher risk profile for complications during serious infection, that is, individuals who are overweight, have high blood pressure, high blood sugar levels, atherosclerosis, or over 50 years of age [71]. Fruitflow^®^ can help reduce platelet hyperactivity which forms a bridge between the immune response to infection and development of thrombotic complications. Intervention with a dietary antiplatelet will not likely overturn serious thrombotic complications such as those observed in sepsis or severe COVID-19, and we are not advocating for its use in such conditions. Rather, we suggest that Fruitflow^®^ may help improve the resilience of our bodies’ reaction to initial or mild infection through its beneficial effects on blood platelets and thrombosis. The safety and suitability of Fruitflow^®^ as a dietary antiplatelet are an important aspect of this suggestion, as its safety profile and relatively mild action render it useable by generally healthy adults [71]. Immunity and resilience to viral infection is a focus area taking centre stage in the public consciousness worldwide. There is a clear drive from the public to find ways of boosting resilience and staying healthy in the face of infectious diseases, and it is imperative that the potential role of nutrition or dietary supplements is investigated.

## 5. Emerging Areas of Interest for Dietary Antiplatelets

### 5.1. Platelet Hyperactivity during Menopause

A growing focus on menopause, and changes in women’s cardiovascular health accompanying the reduction in oestrogen levels during menopause, has highlighted platelet hyperactivity as a target in this area. From puberty onwards, oestrogens play an integral role in the life of a woman. Apart from governing the reproductive system, oestrogens affect mood, appetite and energy. They also confer enviable protection on the cardiovascular system so that women’s risk of CVD is significantly lower than men’s for 50–60 years [72]. When natural levels of oestrogen decline during perimenopause, this protective effect is lost. In the space of 5–10 years, women’s CVD risk equals that of men [73].

In part, this effect relates to the loss of oestrogen signalling to regulate the elasticity of blood vessels. Oestrogens help regulate this dynamic behaviour by signalling for the release of NO in the blood vessels, causing them to dilate [74]. The vasoconstricting effect of the menopausal drop in oestrogens is compounded by an age-related reduction in NO. Oestrogens also directly affect the control of platelet activation in women via platelet oestrogen receptors, which have an anti-aggregant effect when oestrogen is bound. This extra platelet control contributes to reducing women’s cardiovascular risk while oestrogen levels are high [75]. During perimenopause, platelet oestrogen receptors reduce in number, and after menopause, they disappear. Alongside reduced nitric oxide, this removes a layer of protection from women’s platelets. Simultaneously, the dropping oestrogen levels cause increased instability from disturbed metabolism, an unhealthy balance of blood fats, and damaged blood vessels. Platelets can become persistently sticky after menopause, both a cause and consequence of women’s increased cardiovascular risk profile.

Reducing platelet hyperactivity during and after menopause may go some way towards compensating for the loss of the protective effects of oestrogen on the cardiovascular system. Alongside diet and exercise advice, and HRT where appropriate, dietary antiplatelets can be potentially useful adjunct therapy suitable for long term use.

### 5.2. Erectile Dysfunction

Another major consequence of low NO levels and the associated increased platelet hyperactivity in men is erectile dysfunction (ED), a disorder with widespread prevalence [76]. The successful use of Sildenafil in ED highlighted the central role of nitric oxide (NO) in mediating normal erection and in the pathogenesis of ED [77]. Platelet hyperactivity contributes to countering erection by releasing vasoconstrictors, superoxide, and other oxygen free radicals [78]. Low NO facilitates the adhesion of platelets and leukocytes to vascular tissue via P-selectin and GPIIb-IIIa. The adhered platelets then release vasoconstrictors, hindering erection.

Some researchers consider that ED may be a manifestation of cardiovascular disease, and certain similarities in pathology exist. A recent study used the antiplatelet drug aspirin (acetylsalicylic acid) in men with vasculogenic ED and a high mean platelet volume (associated with platelet activation); this study concluded that taking 100 mg aspirin daily significantly reduced the symptoms of ED [79]. However, daily aspirin therapy is associated with enhanced bleeding risks, which raises the question of whether a dietary antiplatelet could be similarly successful. It is already known that diet and lifestyle impact ED and that altering nutritional factors can have beneficial effects [80]. In fact, diets close to the Mediterranean diet profile (including high tomato intake) are associated with a reduced burden of ED [81]. Fruitflow^®^ blocks expression of P-selectin on the platelet surface and prevents GPIIb-IIIa from binding; it has similar effects to low dose aspirin [14]. Thus, we hypothesise that these characteristics, combined with their better safety profile, merit further investigation of Fruitflow^®^ in ED.

### 5.3. Allergic Responses

With links to both inflammation and the immune system via the complement pathway, the allergic response is another area in which platelets are implicated. Platelet-leukocyte aggregates are well documented in allergic responses [82], and investigation of new pathways by which platelets can be suppressed without compromising their primary haemostatic function is an area of active research [83]. While there is little available information on nutritional interventions which help to suppress allergic responses, dietary antiplatelets may be an option worth exploring.

## 6. Conclusions

Platelets have multifaceted functions which generate a complicated set of interactions with other vascular cells, leading to many roles outside haemostasis. As our understanding of the role of platelet activation in response to— and in complicating—inflammatory and infectious illnesses grow, it becomes more apparent that platelet-targeted treatments are necessary outside the field of CVD. Dietary antiplatelets such as Fruitflow^®^ can help provide suitably gentle and safe yet efficacious treatments to improve public health in response to a wide range of health challenges.

## Figures and Tables

**Table 1 nutrients-13-02184-t001:** Post-treadmill exercise effects on platelet activation, coagulation and inflammation in men aged 18–55 (*n* = 6).

	% Increase from Baseline Value Post-Exercise		
	Plasma Microparticle Count ^1^	Plasma Thrombin Generation Capacity ^2^	Circulating IL-6 ^3^
Placebo treatment (*n* = 6)	93 ± 18	120 ± 9	345 ± 32
Fruitflow^®^ treatment (*n* = 6)	21 ± 6 *	13 ± 4 *	145 ± 22 *

^1^ Plasma microparticle count is a measure of platelet aggregability, measured by flow cytometry. ^2^ Plasma thrombin generation capacity is a measure of coagulation and is measured by a fluorescence-based assay in which evolution of thrombin is monitored over time. ^3^ IL-6 was measured by ELISA. Values are given as mean ± SD. Significant differences from placebo treatment are shown by * (*p* < 0.05).

## Data Availability

Not Applicable.

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
