# Peer review of "Dietary Antiplatelets: A New Perspective on the Health Benefits of the Water-Soluble Tomato Concentrate Fruitflow®"

_nutrients, 2021, doi:10.3390/nu13072184_

Round 1

Reviewer 1 Report

I have no more comments for the authors. 

Reviewer 2 Report

Interesting paper well organized and discussed

This manuscript is a resubmission of an earlier submission. The following is a list of the peer review reports and author responses from that submission.

Round 1

Reviewer 1 Report

Journal Name: Nutrients

Title of the Manuscript: Dietary antiplatelets: a new perspective on the health benefits 2 of the water-soluble tomato concentrate Fruitflow®

The authors focus on a potential new uses for dietary antiplatelets outside the field of cardiovascular health, with specific reference to the water-soluble tomato extract Fruitflow®.

This is an interesting scientific hypothesis and well-written review.

However, important and major information is missing

  • At the beginning of each sub-section, it is important to make the link between the in vitro results and real life. To answer this, the authors should be able to carry out real clinical research with randomized studies, taking into account the topics detailed in the review, such as diet, environmental stresses, exercise, hormonal cycle, etc...
  • Moreover, it is important that the authors have results on a large number of subjects with regard to the different conditions defined above and an adequate statistical methodology, especially in terms of statistic power. This is not sufficiently addressed in this review.
  • It is important that the authors can produce a paragraph concerning the transfer of the active products of the water-soluble tomato concentrate Fruitflow, from the digestive system and into the bloodstream. A mechanistic and physiological approach is needed.
  • In addition, the authors should also make a paragraph concerning other nutritional approaches capable of reducing inflammation (in relation to blood platelets), insisting on the physiological mechanisms. This paragraph would allow emphasizing the effects of water-soluble tomato concentrate Fruitflow.
  • Paragraph: "3.1 Exercise-induced inflammation" The authors should more insist on the part of exercise "alone" to limit inflammation and the additional value of balanced nutrition and then the additional value of water-soluble tomato concentrate Fruitflow.
  • Table 1: as described by the authors the data are much too preliminary (n=6) to identify scientific concepts, the number of experiments must be increased and this number must depend on a defined statistical methodology to characterize the statistical power.
  • Paragraph: "3.2. Air pollution", this approach is interesting but as the physical exercise, the authors should more insist on the part of pollution "alone" to limit the inflammation and the value to add of the water-soluble tomato concentrate Fruitflow. Moreover, the individual characteristics of humans (genetic susceptibility) and their environment must be taken into account to validate the scientific concepts concerning the interest of water-soluble tomato concentrate Fruitflow.
  • For both paragraphs 3.1 and 3.2, a description of possible "animal models" concerning inflammation and these two phenomena (exercise and pollution) would allow a better understanding of the mechanistic approach of the phenomena identified by the authors on the decrease of inflammation after ingestion of water-soluble tomato concentrate Fruitflow.
  • 4. Dietary antiplatelets in immunity. I understand the interest of the authors to mention the COVID infection in this review, to make it more updated. However, this whole part should be deleted because it is really out of the message of the review. This paragraph does not add anything to the review, and even deserts it. COVID infection (but also other pathogens) is very complex and the inflammatory response is multifactorial and sometimes cascading. The authors should stop the paragraph after the line 289.
  • 5. Emerging areas of interest for dietary antiplatelets. This paragraph is very interesting but the authors should always put each sub-section in a "nutritional" context. At the beginning of each sub-section, the authors should give the impact of nutrition:
    • in general, on the immune response,
    • the immune response in relation to the physiological dysfunction of interest (menopause, erection, allergy)
    • the immune response targeted on platelet physiology
  • Conclusion section: The authors should suggest more of their own position on other international publication/context studies and not just relate the studies of bibliography. The authors are experts in the field and must take certain risks and clearly state their position, while at the same time referring to contradictory studies. The authors should in the discussion state the limitations of scientific hypothesis and own data.

Minor information is missing

  • Are figures 1 and 2 original?
  • « Dietary antiplatelet come into play under circumstances where persistent platelet activation arises in response to dietary and lifestyle factors – notably smoking or exposure to smoke / air pollution, consistently high levels of plasma glucose, and certain exercise patterns.” Please insert a bibliographic reference

Author Response

Reviewer-1

Journal Name: Nutrients

Title of the Manuscript: Dietary antiplatelets: a new perspective on the health benefits 2 of the water-soluble tomato concentrate Fruitflow®

The authors focus on a potential new uses for dietary antiplatelets outside the field of cardiovascular health, with specific reference to the water-soluble tomato extract Fruitflow®.

This is an interesting scientific hypothesis and well-written review.

Many thanks for your positive comments on our manuscript. We appreciate very much.

However, important and major information is missing 

  • At the beginning of each sub-section, it is important to make the link between the in vitro results and real life. To answer this, the authors should be able to carry out real clinical research with randomized studies, taking into account the topics detailed in the review, such as diet, environmental stresses, exercise, hormonal cycle, etc...
  • Moreover, it is important that the authors have results on a large number of subjects with regard to the different conditions defined above and an adequate statistical methodology, especially in terms of statistic power. This is not sufficiently addressed in this review.
    • As we have previously summarised the studies undertaken on Fruitflow in a review article, we did not consider including a full discussion of the available studies, with numbers of subjects, study designs and statistical treatments.  We have however referenced all the studies.  To try to clarify the extent of the body of work available, we have also expanded on the information in the introduction, from Line 72 - 83.
  • It is important that the authors can produce a paragraph concerning the transfer of the active products of the water-soluble tomato concentrate Fruitflow, from the digestive system and into the bloodstream. A mechanistic and physiological approach is needed.
    • This has been included in brief from Line 68-74.

  In addition, the authors should also make a paragraph concerning other nutritional approaches capable of reducing inflammation (in relation to blood platelets), insisting on the physiological mechanisms. This paragraph would allow emphasizing the effects of water-soluble tomato concentrate Fruitflow.

    • More background information on the use of dietary antiplatelets in general, and the nature of other dietary antiplatelets, has been included as suggested, in a new paragraph from Line 50-63..
  • Paragraph: "3.1 Exercise-induced inflammation" The authors should more insist on the part of exercise "alone" to limit inflammation and the additional value of balanced nutrition and then the additional value of water-soluble tomato concentrate Fruitflow.
    • We have briefly touched on this at the start of this section, Line 159 onwards.
  • Table 1: as described by the authors the data are much too preliminary (n=6) to identify scientific concepts, the number of experiments must be increased and this number must depend on a defined statistical methodology to characterize the statistical power.
    • We have further emphasised the pilot nature of this data and pointed out its limitations more clearly, Line 237 -240.
  • Paragraph: "3.2. Air pollution", this approach is interesting but as the physical exercise, the authors should more insist on the part of pollution "alone" to limit the inflammation and the value to add of the water-soluble tomato concentrate Fruitflow. Moreover, the individual characteristics of humans (genetic susceptibility) and their environment must be taken into account to validate the scientific concepts concerning the interest of water-soluble tomato concentrate Fruitflow.
    • We have included some remarks about interactions with genes / environments to limit our hypothesis as indicated, Line 303-305.
  • For both paragraphs 3.1 and 3.2, a description of possible "animal models" concerning inflammation and these two phenomena (exercise and pollution) would allow a better understanding of the mechanistic approach of the phenomena identified by the authors on the decrease of inflammation after ingestion of water-soluble tomato concentrate Fruitflow.
    • As far as platelets are concerned, animal models are of limited use only – animal platelets (rats, mice, dogs, horses) are different to human platelets in fundamental ways.  For example rat platelets have entirely different ADP receptors.  Thus we prefer to use cell models and human platelets only, as otherwise conclusions are difficult to generalise.
  • 4. Dietary antiplatelets in immunity. I understand the interest of the authors to mention the COVID infection in this review, to make it more updated. However, this whole part should be deleted because it is really out of the message of the review. This paragraph does not add anything to the review, and even deserts it. COVID infection (but also other pathogens) is very complex and the inflammatory response is multifactorial and sometimes cascading. The authors should stop the paragraph after the line 289.
    • In this case, we have combined the input of Reviewer 2.  We have emphasised the complexity of the situation, and included details of treatments for other infections, rather than eliminating what is a very interesting and topical point.  The complexity of the inflammatory response to infection is acknowledged in the paragraph, and we feel that there is justification for exploring the potential role of nutrients in modulating this response.
  • 5. Emerging areas of interest for dietary antiplatelets. This paragraph is very interesting but the authors should always put each sub-section in a "nutritional" context. At the beginning of each sub-section, the authors should give the impact of nutrition:
    • in general, on the immune response,
    • the immune response in relation to the physiological dysfunction of interest (menopause, erection, allergy)
    • the immune response targeted on platelet physiology
      • We are not very clear on what is required here, especially with regard to the comments on immune response ?  We have mentioned the importance of nutrition in menopause on Line 415-416, and have highlighted that nutritional intervention can alter ED pathology on Line 433-436, and we hope this is in line with the comments above.
  • Conclusion section: The authors should suggest more of their own position on other international publication/context studies and not just relate the studies of bibliography. The authors are experts in the field and must take certain risks and clearly state their position, while at the same time referring to contradictory studies. The authors should in the discussion state the limitations of scientific hypothesis and own data.
    • Our intention is to highlight the possibility that dietary antiplatelets may have application in areas outside cardiovascular health.  As many of these applications are not currently widely explored, we are not aware of any contradictory studies which we could reference here.  We have clearly stated that our own data is preliminary and is included really for illustration of potential new uses of antiplatelets, rather than as proof of efficacy. We hope the concluding statements we have included come across clearly as they do represent our position in support of the idea that antiplatelets have more uses than currently appreciated.

  Minor information is missing

 Are figures 1 and 2 original?

    • Yes, they are.
  • « Dietary antiplatelet come into play under circumstances where persistent platelet activation arises in response to dietary and lifestyle factors – notably smoking or exposure to smoke / air pollution, consistently high levels of plasma glucose, and certain exercise patterns.” Please insert a bibliographic reference
    • We have inserted 4 references here.

Reviewer 2 Report

Based on the critical role of platelet aggregation and hyperactivation in several immune functions and tissue damage, O'Kennedy et al are reviewing on the potential benefits of utilizing dietary antiplatelets and especially Fruitflow in inflammation, chronic diseases and some infections. Authors also expand the perspectives of such use in various other conditions and especially menopause, erectile dysfundtion and allergy. 

The review is of interest, is nicely writen and well-organised in separate sections. 

I have some comments:

 - Figure 1

There are no signs on SDs or SE in the bars on the graphs, which may imply that the experiments have been done once without replication.  If this is the case you should mention it, otherwise the SDs should be added on the graphs.  

Line 179, (label of the Figure): You should define what was the control for this experiment. The same for section B of this Figure

- On part 4 (Lines 309-324) the authors refer to the potential benefit of antiplatelet therapies in SARS-CoV2 infection. It would be interesting to mention relevant benefits reported from the use of antiplatelet therapies and especially dietary antiplatelets on previous relevant virus infectious (SARS-CoV, MERS-CoV etc)

 - Line 324: Correct Sars-CoV2 to SARS-CoV2

 - "Allergic responses" part refers to 5.3 not 5.2. 

Author Response

Reviewer-2

Comments and Suggestions for Authors

Based on the critical role of platelet aggregation and hyperactivation in several immune functions and tissue damage, O'Kennedy et al are reviewing on the potential benefits of utilizing dietary antiplatelets and especially Fruitflow in inflammation, chronic diseases and some infections. Authors also expand the perspectives of such use in various other conditions and especially menopause, erectile dysfundtion and allergy. 

The review is of interest, is nicely writen and well-organised in separate sections. 

Many thanks for your positive comments on our manuscript. We appreciate very much.

I have some comments:

 - Figure 1

There are no signs on SDs or SE in the bars on the graphs, which may imply that the experiments have been done once without replication.  If this is the case you should mention it, otherwise the SDs should be added on the graphs.  

            This has now been amended.

Line 179, (label of the Figure): You should define what was the control for this experiment. The same for section B of this Figure

            The control is identified on Line 218 (saline).

- On part 4 (Lines 309-324) the authors refer to the potential benefit of antiplatelet therapies in SARS-CoV2 infection. It would be interesting to mention relevant benefits reported from the use of antiplatelet therapies and especially dietary antiplatelets on previous relevant virus infectious (SARS-CoV, MERS-CoV etc)

            We have mentioned the use of antiplatelet medications in influenza, which is an intervention sometimes used when pneumonia develops.  However we are not aware of any use of dietary antiplatelets in either influenza or Mers-CoV.  We have referenced the recent publications which refer to potential for dietary antiplatelets in the current SARS-CoV epidemic.  The potential impact of the platelet in infection has only recently really come to the forefront of attention, and so although there are some studies currently in the planning stage, we do not have much to reference here yet.

 - Line 324: Correct Sars-CoV2 to SARS-CoV2

            Completed

 - "Allergic responses" part refers to 5.3 not 5.2. 

            Completed

Round 2

Reviewer 1 Report

Thanks for the comments and new input regarding this version. However, I hope you could response more to my comment about the following part:

  • For both paragraphs 3.1 and 3.2, a description of possible "animal models" concerning inflammation and these two phenomena (exercise and pollution) would allow a better understanding of the mechanistic approach of the phenomena identified by the authors on the decrease of inflammation after ingestion of water-soluble tomato concentrate Fruitflow.
    • As far as platelets are concerned, animal models are of limited use only – animal platelets (rats, mice, dogs, horses) are different to human platelets in fundamental ways.  For example rat platelets have entirely different ADP receptors.  Thus we prefer to use cell models and human platelets only, as otherwise conclusions are difficult to generalise.

Thanks for considering my comment in a new version, I think this is important in the context of this review.

Author Response

Editor,

Nutrients (ISSN 2072-6643)

Re:  Manuscript ID: nutrients-1187641

Title: Dietary antiplatelets: a new perspective on the health benefits of the water-soluble tomato concentrate Fruitflow® by Niamh O’Kennedy , Ruedi Duss , Asim K Duttaroy *

Dear Anne,

We appreciate very much the reviewer-1’s comments. We have now modified the manuscript to accommodate the comments.

We hope the revised manuscript is now acceptable fro publication in your journal.

With Regards,

Asim K. Duttaroy

Reviewer 1

Thanks for the comments and new input regarding this version. However, I hope you could response more to my comment about the following part:

  • For both paragraphs 3.1 and 3.2, a description of possible "animal models" concerning inflammation and these two phenomena (exercise and pollution) would allow a better understanding of the mechanistic approach of the phenomena identified by the authors on the decrease of inflammation after ingestion of water-soluble tomato concentrate Fruitflow.
    • As far as platelets are concerned, animal models are of limited use only – animal platelets (rats, mice, dogs, horses) are different to human platelets in fundamental ways.  For example rat platelets have entirely different ADP receptors.  Thus we prefer to use cell models and human platelets only, as otherwise conclusions are difficult to generalise.

Thanks for considering my comment in a new version, I think this is important in the context of this review. 

For exercise-induced platelet activation, we remain of the belief that animal models may not provide clearly generalisable data; however, we do recognise that some of the available mouse models may indeed be useful in tracking the extent of induced inflammation, and may allow exploration of specific conditions (for example exercising under hypoxia) more easily than human studies.  We have therefore inserted a line to this effect (Lines 240-243, highlighted in blue in the amended manuscript).  In the context of air pollution, we are aware of a range of established animal models which can be used to examine the overall cardiovascular impact of exposure to particulate matter; we have inserted a comment to reflect that such models may form a useful next step in evaluating the impact of Fruitflow on the longer-term effects of air pollution (Lines 308-312, highlighted in blue).
